# Characterization of the Biological Activities of a New Polyphenol-Rich Extract from Cinnamon Bark on a Probiotic Consortium and Its Action after Enzymatic and Microbial Fermentation on Colorectal Cell Lines

**DOI:** 10.3390/foods11203202

**Published:** 2022-10-14

**Authors:** Alessandra De Giani, Stefania Pagliari, Jessica Zampolli, Matilde Forcella, Paola Fusi, Ilaria Bruni, Luca Campone, Patrizia Di Gennaro

**Affiliations:** Department of Biotechnology and Biosciences, University of Milano-Bicocca, 20126 Milano, Italy

**Keywords:** cinnamon, antimicrobial activity, in vitro digestion, probiotics, functional foods, nutraceuticals

## Abstract

Cinnamon polyphenols are known as health-promoting agents. However, their positive impact depends on the extraction method and their bioaccessibility after digestion. In this work, cinnamon bark polyphenols were extracted in hot water and subjected to an in vitro enzymatic digestion. After a preliminary characterization of total polyphenols and flavonoids (respectively 520.05 ± 17.43 µgGAeq/mg and 294.77 ± 19.83 µgCATeq/mg powder extract), the extract antimicrobial activity was evidenced only against *Staphylococcus aureus* and *Bacillus subtilis* displaying a minimum inhibition growth concentration value of 2 and 1.3 mg/mL, respectively, although it was lost after in vitro extract digestion. The prebiotic potential was evaluated on probiotic *Lactobacillus* and *Bifidobacterium* strains highlighting a high growth on the in vitro digested cinnamon bark extract (up to 4 × 10^8^ CFU/mL). Thus, the produced SCFAs and other secondary metabolites were extracted from the broth cultures and determined via GC-MSD analyses. The viability of healthy and tumor colorectal cell lines (CCD841 and SW480) was assayed after the exposition at two different concentrations (23 and 46 µgGAeq/mL) of the cinnamon extract, its digested, and the secondary metabolites produced in presence of cinnamon extract or its digested, showing positive protective effects against a tumorigenic condition.

## 1. Introduction

Nowadays, the interest versus nutraceuticals is progressively increasing due to nutritional and therapeutic potentials linked to health amelioration, delay senescence, diseases prevention, and supporting the proper functioning of the human body [1]. Consequently, there is an increase of attention to nutraceuticals, including polyphenols, as possible new dietary supplements [2]. Indeed, they are recognized for the modulation of positive effects on humans, reducing the inflammation status and ameliorating the antioxidant potential that decreases the oxidative stress rate [2,3]. The term “polyphenols” encompasses several families of molecules, including phenolic acids, flavonoids, catechins, hydroxycinnamates, coumarins, anthocyanins, ellagic acid, lignans, ellagitannins, and isoflavones [2]. Several polyphenolic compounds are also known for their antimicrobial potential and are approved as GRAS (Generally Recognized As Safe) substrates for food products [4]. Various botanicals are natural sources of polyphenols, for example, thyme (*Thymus vulgaris* L.), tea (*Camellia sinensis* L.), garlic (*Allium sativum* L.), turmeric (*Curcuma longa* L.), and cinnamon (several species belonging to the *Cinnamomum* genus). Nevertheless, particular interest is raised by the *Cinnamomum* genus (Lauraceae family) because it groups more than three hundred evergreen aromatic trees and shrubs [5]. Among them, *Cinnamon zeylanicum* Blume (a synonym of *Cinnamon verum* J. Presl, known as Sri Lanka cinnamon), *Cinnamon loureiroi* Nees (Vietnamese variety), *Cinnamon burmanni* (Nees & T. Nees) Blume (Indonesian cinnamon), and *Cinnamon aromaticum* Nees (or *Cinnamon cassia* L. J. Presl, known as Chinese cinnamon) have significant economic importance, because they are used in the culinary field worldwide [6] but also as health-promoting agents. Indeed, they are employed in lowering the inflammation state, gastrointestinal disorders, and infections of urinary tract due to their antimicrobial features [4,7]. In *C. zeylanicusm* (or *C. verum*), the main isolated and identified compounds are phenols, like vanillic, caffeic, *p*-coumaric, ferulic acids, and the volatile ones. The bark essential oil is characterized by the presence of cinnamaldehyde, whose content varies from 62–73% to 90% depending on the method of extraction [4]. Cinnamaldehyde and cinnamic acid revealed to have a role in the modulation of certain cellular pathways, improving the glucose balance in vivo [7]. However, clinical studies highlight that the form in which cinnamon is provided could be critical because different extract preparations (employing water-based and/or organic solvents) and powders would yield different bioavailability levels correlated to distinct phytochemical compositions [7]. Furthermore, an important role in the bioaccessibility (i.e., the fraction of the total amount of a compound that is eventually available for absorption) of the beneficial polyphenols is played by their digestion by both human enzymes and those encoded by the bacteria that make up the gut microbiota [8]. Indeed, phenolic acids are linked to lignin by ether linkages of the plant material [9]. Nevertheless, gastric acidity could make the phenolic compounds more stable and promote their release from the natural matrix, favoring the survival in the stomach. Instead, the mild alkaline pH of the intestine could damage the phenolic acids. As a consequence, virtually up to 95% of dietary polyphenols are not absorbed by the small intestine, thus arrived in the colon [10], where the intestinal bacteria develop de-glycosylating activities, releasing aglycones then degraded to simpler phenolic by-products, as hydroxyphenyl acetic acid [11,12]. The final derivative or aglycon can be absorbed at the enterocyte level, where they are submitted to phase II metabolism (methylation, sulfonation, and glucuronidation), and enter the bloodstream [12].

In this scenario, the effectiveness of dietary polyphenols is probably due to the phenolic metabolites derived from the gut microbiota intervention, instead of the original forms found in food [13]. For example, hydroxycinnamic acids are commonly esterified to sugars, organic acids, and lipids [12], but the esterase capacity of lactic acid bacteria is released during fermentation. Then, the hydroxycinnamic acids are converted via decarboxylation and reduction reactions by the lactic acid metabolism [14]. Under in vitro conditions, it was shown that anthocyanins and phenolic compounds could be transformed into short-chain fatty acids (SCFAs), because of the stimulation of the growth of *Lactobacillus* and *Bifidobacterium* species [9]. Acetic, propionic, and butyric acids are included in the finest SCFAs, known for the maintenance of correct body functions. In particular, butyric acid is the main energy for colonocytes [9]. However, the metabolic processes related to phenolic acids are strictly dependent on the lactic acid bacterium and poorly characterized [14]. Nevertheless, for example, phenolic acid esterases and decarboxylases are encoded in the *Lactiplantibacillus plantarum* genome [15,16,17,18] and this is probably connected to the fact that dietary polyphenols poorly inhibit the growth of the beneficial Lactobacilli [19]. Therefore, to predict the stability, the bioavailability, and then the effects of polyphenols or natural extract containing them, in vitro digestion can represent a reliable model mimicking in vivo conditions [20]. The models could include enzymatic and bacterial digestions [9,21,22]. A general standardized static digestion method was used within the COST INFOGEST network. It mimics physiologically relevant conditions, including oral, gastric, and small intestinal digestions; therefore, it is based on the presence of digestive enzymes, taking into account their real contents, working pH and temperature, digestion time, and salt concentrations [23]. Regarding bacterial digestion, several in vitro models are described in the literature, differing in complexity [24].

In this context, the idea of this work is to evaluate the potential of cinnamon bark in promoting healthy beneficial effects due to its polyphenolic components. However, the extraction method plays a key role to recover the bioactive molecules and the bioaccessibility can be impaired by oral, gastric, and intestinal digestion. Furthermore, also the intestinal microbiota has a prominent role on the bioavailability, thus positively impacting on the host. In order to reach this aim, the phenolic components were extracted from *Cinnamomum verum* J. Presl (sin. *C*. *zeylanicum* Blume) bark and they were characterized for their effects versus beneficial bacteria before and after an in vitro enzymatic digestion. Furthermore, the potential beneficial effects on human colorectal cell lines of both the extracts (before and after an enzymatic digestion) and the effects mediated by the secondary metabolites produced by the probiotic bacteria growth on the cinnamon extract was evaluated.

## 2. Materials and Methods

### 2.1. Extraction of Cinnamon Bark Bioactives

Cinnamon bark was provided by Epo S.r.l. (Milano, Italy). It was stored at 25 °C away from heat and light sources until its usage. The thinly blending of the natural source was obtained through rounds of 2 min at 6,000 rpm with a knife mill (Grindomix GM-200, Restek GmbH, Haan, Germany), preventing the matrix heating. Phenolic-based molecules were recovered from cinnamon bark by maceration, as described in Cheng et al. [7] with minor adjustments. 1 g of milled plant material was placed for 1 h at 60 °C under constant agitation, using three different extraction solvents: phosphate buffer (PB), 70% EtOH, and water with a matrix:solvent ratio of 1:20. Then, the obtained speciments were centrifuged at 600 rpm for 15 min. Subsequently, the supernatants were quantitatively recovered and mixed with 75% EtOH (1:1 *v*/*v*) for 1 h at 4 °C allowing the concentration of the polysaccharide fraction. Finally, the solution was filtrated (Whatman No. 1 filter), the organic solvent removed at 40 °C through Rotavapor (model Strike-300, Steroglass Italia S.r.l, San Martino in Campo, Italy), and freeze-dried (Alpha 1-2 LD, Christ, Heidelberg, Germany). Extraction yields were determined gravimetrically and expressed as mg/g of powder extract.

### 2.2. Qualitative and Quantitative Analysis of Cinnamon Bark Extract

Qualitative analysis of the cinnamon bark extract principal compounds was completed using HPLC-UV equipment. The analysis was performed using a Zorbax SB-C18 analytical column (4.6 × 250 mm I.D., 5 µm) held at 30 °C for all chromatographic runs, and UV traces were recorded at 280 nm. The mobile phases were water (A) and methanol (B), both with 0.1% formic acid. Injections were performed using the following gradient: 0–2 min, 5% B; 5–7 min, 5–10% B; 7–10 min, 10–25% B; 10–13 min, 25% B; 13–15 min, 25–40% B; 15–20 min, 40–70% B; 20–25 min, 70–85% B. The column was flushed with 95% B for 5 min at the end of every injection. Then, it was re-equilibrated for 5 min before the next run. A flow rate of 1 mL/min and an injection volume of 10 μL was used. 

Quantitative analyses of total polyphenols, flavonoids, and catechins in the cinnamon bark extract were carried out through different spectroscopic methods. For the quantification of total polyphenols, the Folin–Ciocalteu method was used as reported by Campone et al. [25] with modifications. Gallic Acid (GA) (Sigma-Aldrich, Milano, Italy) was used for the calibration curve in a concentration ranging from 0 to 100 μg/mL. The assay was set up in a cuvette using 400 μL of water, 80 μL of the sample or GA solution, 40 μL of Folin–Ciocalteu reagent, and finally 480 μL of a 10.75% Na_2_CO_3_. The absorbance values were recorded at 760 nm after 30 min of incubation. Results were reported as µgGAeq/mg of powder extract.

To quantify the total flavonoid content (TFC), an aluminum chloride assay was used as reported by Zhi-Shen et al. [26] with some modifications. Catechin (CA) (Sigma-Aldrich, Milano, Italy) was used for a calibration curve in quantities between 0 and 14 µg/mL. The assay was set up in a cuvette using 400 μL of water, 340 μL of the sample or CA solution, 30 μL of a 5% sodium nitrate, 30 μL of a 10% aluminum chloride solution, and 200 μL of 1 N sodium hydroxide. After each step, the sample was incubated for 5 min at room temperature and when the last reagent was added, it was read at 510 nm. Obtained results were expressed as µgCATeq/mg of powder extract.

To quantify the amount of catechin, the flavan-3-oil assay was used as reported by McMurrough and McDowell [27] with some modifications. Catechin (CA) (Sigma-Aldrich, Milano, Italy) was used for a calibration curve in a concentration between 0 and 50 µg/mL. The assay was set up in a cuvette using 100 μL of the sample or CA solution and 500 μL of DMAC. After an incubation of 2 min at room temperature, 500 μL of water was added and the absorbance was read at the wavelength of 640 nm. Quantities were expressed as µgCATeq/mg of powder extract.

### 2.3. In Vitro Oral and Gastrointestinal Digestion Employing INFOGEST Protocol

The simulation of the digestion was performed using the INFOGEST protocol [23,28] as shown in Figure 1. At the end of step 3, the aliquot was immediately acidified to pH 3 with 6 N HCl. In this condition the further degradation of the polyphenols was stopped and at the same time the enzymes precipitate and subsequently pulled out at 14,000 rpm for 10 min at 4 °C.

### 2.4. Bacterial Strains and Culture Conditions

Microbial strains are the same employed in the previously published papers [29,30,31]: *Lactobacillus acidophilus* LMG P-29512 (formerly DSM 24936), *Lactiplantibacillus plantarum* DSM 24937 (formerly *Lactobacillus plantarum* DSM 24937), *Lacticaseibacillus rhamnosus* LMG P-29513 (formerly *Lactobacillus rhamnosus* DSM 25568), *Limosilactobacillus fermentum* DSM 25176 (formerly *Lactobacillus fermentum* DSM 25176), *Limosilactobacillus reuteri* DSM 25175 (formerly *Lactobacillus reuteri* DSM 25175), *Bifidobacterium animalis* subsp. *lactis* LMG P-29510 (formerly DSM 25566), *Bifidobacterium longum* subsp. *longum* DSM 25174, *Bifidobacterium longum* subsp. *infantis* LMG P-29639. The *Lactobacillus* and *Bifidobacterium* strains are supplied by a private collection of the company Roelmi HPC (Origgio, Italy). They were previously selected for the probiotic features and the characterization is reported in previous works [29,30,31]. The other strains from ATCC collection *Escherichia coli* ATCC 25922, *Staphylococcus aureus* ATCC 6538, *Pseudomonas aeruginosa* ATCC 9027, *Bacillus subtilis* ATCC 6633, and *Candida albicans* ATCC 10231 were used as potential pathogen strains.

Probiotic bacteria were activated by growing them as reported by De Giani et al. [31].

For further growth trials in presence of the polyphenol-rich cinnamon bark extract and the in vitro digested polyphenol-rich cinnamon bark extract, a modified MRS (mMRS, [32]), in absence of glucose and added with 0.03% *w*/*v* L-cysteine was employed. Cinnamon bark extract (rich in polyphenols) and the in vitro digested form of the same extract were added to mMRS as the sole source of growth at a concentration of 1% *w*/*v*. In both cases, the substrates were sterilized by filtration (MillexSyringe Filter Units, pore size 0.45 μm; Merck Millipore, Darmstadt, Germany) and supplemented in sterility to autoclaved mMRS medium.

The potential pathogen strains were grown in a Luria-Bertani medium adjusted by Lennox [33]. *E. coli*, *P. aeruginosa*, and *S. aureus* were grown at 37 °C in aerobiosis, while *B. subtilis* and *C. albicans* were kept at 30 °C in aerobic conditions.

### 2.5. Screening of the Antimicrobial Activity of Polyphenol-Rich Cinnamon Bark Extract and In Vitro Digested Polyphenol-Rich Cinnamon Bark Extract

The agar well diffusion assay (AWDA) [34] was applied for the investigation of the possible antibacterial potential of the cinnamon bark extract and its in vitro digested form. The tested samples were weighted and dissolved in water, then filter sterilized as previously described. The final concentration was 2% *w*/*v*. The antagonist strains were grown until Optical Density at 600 nm (OD_600nm_) of 0.5, equivalent to approx. 10^7^ CFU/mL. 2.5% *v*/*v* of each bacterial or fungal microorganism was put into 20 mL of LD agar, allowing the solidification. A total of 100 μL of polyphenol-rich cinnamon bark extract or in vitro digested polyphenol-rich cinnamon bark extract were dispensed into each generated well; 100 μL of water worked as a control. The plates dispensed with the different cinnamon samples, or the control were incubated overnight at 30 °C or 37 °C in aerobiosis, depending on the inoculated antagonist, for 24 h. The resultant antimicrobial effect is given by the measurement of the inhibition halo near the well corresponding to the failure of bacterial growth.

Then, the Minimum Inhibitory Concentration (MIC) of the sample that had an antimicrobial effect at 2% *w*/*v* was established by a serial dilution with a factor of 1/5 in Milli-Q water, using the microorganism that resulted sensible to the AWDA. For each growth test, developed in a 96-multiwell (SPL Lifesciences, Pocheon-si, Korea) with a final volume of 100 μL for each well, 10 μL of bacterial cells (10^7^ CFU/mL) were inoculated in LD medium plus the extract to test. After incubation for 24 h at 37 °C or 30 °C, in aerobic conditions, the final OD_600nm_ was measured in a microplate reader (Victor, ELx 800, Milano, Italy).

### 2.6. Growth Experiment with Single Probiotic Strains on Polyphenol-Rich Cinnamon Bark Extract and In Vitro Digested Polyphenol-Rich Cinnamon-Bark Extract

Polyphenol-rich cinnamon bark extract or the in vitro digested polyphenol-rich cinnamon bark extract were prepared as previously described and then added to the sterile mMRS at 1% *w*/*v*. 1 mL of mMRS or mMRS + cinnamon extract (as it is or digested) filled a sterile 24 multiwell (SPL Lifesciences, Pocheon-si, Korea). Then, a correct volume of probiotic pre-inoculum was put into each well, to reach an OD_600nm_ of 0.1, corresponding to 3 × 10^7^ CFU/mL. Then, the microtiters were capped and placed at 37 °C for 48 h. At the end of the experiment, the growth of each probiotic strain was evaluated as CFU/mL, plating three consecutive dilutions on MRS plates of a serial dilution in base 10. Triplicates of each experiment were conducted.

### 2.7. Assessment of Biotranforming Capacities of Single Probiotic Strains and Combined as a Probiotic Consortium

The potentially released biotransformed forms of polyphenols and SCFAs after polyphenol-rich cinnamon bark extract and its in vitro digested form were extracted and analyzed in GC-MSD as reported by De Giani et al. [31]. These experiments were conducted with *L. plantarum* and *B. animalis* subsp. *lactis* as examples of Lactobacilli and Bifidobacteria, respectively; and using all the mixed probiotic strains as a probiotic consortium. The consortium was prepared as described by De Giani et al. [31].

### 2.8. Maintenance of Cell Lines

CCD841 (ATCC CRL-1790TM) is a healthy mucosa cell line propagated in EMEM medium as reported in De Giani et al. [31]. SW480 (ATCC CCL-228), colorectal tumoral cells, were grown in RPMI 1640 medium supplemented with heat-inactivated 10% fetal bovine serum (FBS), 2 mM L-glutamine, 100 U/mL penicillin, 100 µg/mL streptomycin and maintained at 37 °C in a humidified 5% CO_2_ incubator. Both ATCC cell lines were validated using short tandem repeat profiles as described by De Giani et al. [31].

The employed reagents were provided by EuroClone (EuroClone S.p.A, Pero, Italy).

### 2.9. Cell Viability Assay

Cell viability after exposure to polyphenol-rich cinnamon bark extract and to its in vitro digested form, as well as after the exposure to probiotic secondary metabolites, was evaluated through MTT assay (Sigma-Aldrich, Milano, Italy), in line with the manufacturer’s protocols. 1 × 10^4^ cells/well were seeded and then let grow in a complete medium without phenol red at 37 °C for 24 h in presence of different concentrations of the samples of interest. Then, 10 µL of MTT solution was added to each well. Absorbance at 570 nm upon solubilization after 4 h of incubation was measured using a microplate reader (Victor, ELx 800, Milano, Italy).

### 2.10. Statistical Analysis

All the experiments regarding the growth of probiotic strains have been tripled. Results were exhibited as means ± standard error (SE). The statistically significant differences were assessed by Student’s *t*-test, indicating * *p*-value < 0.1, ** < 0.05 or *** < 0.01.

Regarding the experiments with the healthy and tumor colorectal cell lines, all the experiments were performed in triplicate. Resulted values were depicted as % of mean vitality ± SE. The employed statistic was Dunnett’s multiple comparisons test. The significance was defined as * < 0.1, ** < 0.05 or *** < 0.01.

## 3. Results

### 3.1. Selection of Extraction Method from Cinnamon Bark

The initial experiments were developed to identify the best extraction method, considering the efficiency of the process and the suitability of the obtained extract for further experiments. Three different solvents were considered for bioactive compound recovery from the cinnamon bark. The selected solvents were a hydroalcoholic solution (70% EtOH), water, and 0.02 M PB, pH 7. Extraction yields (EY) achieved for PB, 70% EtOH, and water, were about 10 mg/g, 112 mg/g, 69 mg/g, respectively. Phosphate buffer extraction provided a very low EY; therefore, it was not considered for further steps. The hydroalcoholic extract, although it gave the highest EY, produced an extract with low solubility during microbiological assays conditions, whereas the aqueous extract was completely water-soluble with a satisfactory EY, and therefore was considered a good compromise and selected for further steps.

### 3.2. Preliminary Characterization of the Polyphenol-Rich Cinnamon Bark Extract

To chemically characterized the aqueous extract, a HPLC analysis was carried out. The trace acquired at 280 nm shows a typical chromatographic profile, called hump, that suggested the presence of catechins in the extract [35]. The presence of catechins is subsequently confirmed by spectrophotometric analysis. Moreover, Figure 2A shows the presence of another main peak at retention time of 20.2 min. This analyte was identified as cinnamic acid in comparison with commercial standard, in agreement with literature data [36,37]. Afterwards, the main classes of metabolites (polyphenols, flavonoids, and catechins) were quantified using spectrophotometric methods. The results showed a total polyphenol content of 520.05 ± 17.43 µgGAeq/mg of powder extract, a flavonoids content of 294.77 ± 19.83 µgCATeq/mg of powder extract, and 77.92 ± 3.39 µgCATeq/mg of powder extract as catechins. Due to the high presence of polyphenols, the aqueous cinnamon extract is now called as “polyphenol-rich cinnamon bark extract”.

### 3.3. In Vitro Enzymatic Digestion of Polyphenol-Rich Cinnamon Bark Extract and Characterization of the Digest

In order to verify the real impact of the polyphenol-rich cinnamon bark extract on the probiotic bacteria, the in vitro gastrointestinal digestion process was simulated using INFOGEST protocol. Indeed, INFOGEST has been used in bioaccessibility studies of phytochemicals such as polyphenols and carotenoids, as well as to observe the digestive fate of proteins, lipids, and carbohydrates [23,28]. For the simulated in vitro digestion, the necessary enzymes were added maintaining the correct relationship with the food and the salts present at the level of each phase. The digestion was also performed by mimicking the human body condition such as, peristaltic movements, the temperature, and the pH values in different region of gastrointestinal tract. The contents of the main metabolites were determined at the end of digestive process through spectrophotometric assays. The quantitative results show a reduction of total polyphenols equal to about 50% and almost a total degradation of flavonoids and catechins due to the digestion process (Table 1). This trend is also confirmed by the chromatographic profile (Figure 2B) where the characteristic shape of a broad hump in the base-line produced by catechins disappears and only cinnamic acid remains clearly visible.

### 3.4. Antimicrobial Activity of Polyphenol-Rich and Digested Cinnamon Bark Extract

The antimicrobial activity of the polyphenol-rich cinnamon bark extract and of the in vitro digested polyphenol-rich cinnamon bark extract was evaluated at the concentration of 2% *w*/*v* through AWDA. An inhibition halo was generated if the extract had an antimicrobial effect. The tested strains were the Gram-negative *E. coli* ATCC 25922, and *P. aeruginosa* ATCC 9027; the Gram-positive *S. aureus* ATCC 6538, and *B. subtilis* ATCC 6633; and the yeast *C. albicans* ATCC 10231. As shown by Figure 3 (depicting only the positive results), the antimicrobial action is evidenced only by the polyphenol-rich extract and only against the Gram-positive strains, showing an inhibition halo with a diameter of 1.7 cm against *S. aureus*, and of 1 cm against *B. subtilis*. Therefore, the MIC value for *S. aureus* is 0.2% *w*/*v* (2 mg/mL), while it is 0.13% *w*/*v* (1.3 mg/mL) for *B. subtilis*.

### 3.5. Growth Experiments of Probiotic Bacteria in Presence of the Polyphenol-Rich and Digested Cinnamon Bark Extract

Another beneficial effect of natural extracts is linked to the growth stimulation of helpful bacteria, such as probiotics. Therefore, an in vitro growth assay was developed testing the cinnamon bark extracts (digested or not digested) at 1% *w*/*v* employing *Bifidobacterium* and *Lactobacillus* strains [29]. The inoculated bacteria were 3 × 10^7^ CFU/mL; then the final CFU/mL were recorded after 48 h of anaerobic fermentation. The control condition was the medium containing all the components except the extracts. As shown in Figure 4, almost all strains tolerate the presence of phenolic compounds present in the not digested extract, with a growth value similar to those recorded on the control medium, except for *L. rhamnosus* and *B. longum* subsp. *infantis*, whose growths were impaired, albeit not in a statistically significant way. Interestingly, among the Lactobacilli, *L. reuteri* and *L. fermentum* reached both the highest growth values of 6 × 10^8^ CFU/mL (*vs.* mMRS, *p*-value < 0.1 and < 0.05, respectively) (Figure 4). However, the extract cannot be considered a potential prebiotic.

On the contrary, the digested polyphenol-rich cinnamon bark extract appears to promote the growth of the tested probiotic bacteria. Indeed, the strains reached at least a value of 2 × 10^8^ CFU/mL (Figure 4). Interestingly, *L. plantarum* (<0.05), *L. acidophilus* (<0.1), *L. fermentum* (<0.05), and *B. animalis* subsp. *lactis* (<0.01) showed the most positive responses, even reaching a growth value of 4 × 10^8^ CFU/mL.

### 3.6. Extraction and Characterization of Lactobacillus and Bifidobacterium Secondary Metabolites in Presence of Cinnamic Acid, Polyphenol-Rich, and Digested Cinnamon Bark Extract

In literature is reported that the microbiota transformation of ingested polyphenols mediates their positive outcomes on human health [11]. In particular, the *L. plantarum* species is noteworthy because it seems to be the most specialized among the Lactobacilli in biotransforming the polyphenolic compounds present in vegetable matrices, such as cinnamon extract [14,16].

Therefore, to investigate which compounds are produced by *L. plantarum*, *B. animalis* subsp. *lactis* as examples of Lactobacilli and Bifidobacteria, and by the probiotic consortium (composed of both Lactobacilli and Bifidobacteria), first the probiotics were exposed to *trans*-cinnamic acid because it appears to be the principal component of both polyphenol-rich cinnamon bark extract, and its digested form (Figure 2). Therefore, after 48 h of anaerobic fermentation, the broth cultures were analyzed by liquid–liquid extraction with ethyl-acetate. Then, gas-chromatography analyses were performed, and the chromatograms were compared to the NIST library and to the spectrum of the single compound, *trans*-cinnamic acid. 

Subsequently, the same analyses were carried out after the exposition to polyphenol-rich cinnamon bark extract or to its in vitro digested form.

#### 3.6.1. Analysis of the Metabolites Produced after Biotransformation of *trans*-Cinnamic Acid

From the analyses of the collected broth cultures, *L. plantarum* has the highest biotransforming potential, as shown by the chromatograms in Figure 5A–C. Nevertheless, all the strains also combined as a probiotic consortium could metabolize the acid (Figure 5C). Indeed, in presence of *trans*-cinnamic acid, at a retention time (R_t_) of 16.5 min, after the fermentation of the probiotics, it was possible to detect hydrocinnamic acid which is the biotransformed form of cinnamic acid (Figure 5). Moreover, in the specific chromatogram corresponding to the analysis of *L. plantarum* broth culture, it was possible to detect also pentanoic acid (R_t_ of 11.6 min) and butyric acid (R_t_ of 13.8 min) (Figure 5A) known to be SCFAs, reflecting the slight growth capacity of the strain on the substrate.

#### 3.6.2. Analysis of the Metabolites Produced after Biotransformation of Polyphenol-Rich Cinnamon Bark Extract

With respect to the control condition, samples from fermented polyphenol-rich cinnamon bark extract present a different peak profile (Figure 6). Indeed, the chromatogram of the negative control (i.e., the ethyl-acetate extracted polyphenol-rich cinnamon bark extract dissolved in mMRS medium) presented only one compound, assigned to citric acid, that is a component of the mMRS medium (Figure 6A). Interestingly, the compound is no longer detected in the presence of the individual probiotic bacteria. Regarding *L. plantarum* (Figure 6B), the first most abundant peak (R_t_ of 8.2 min) is associated with lactic acid, the most produced metabolite from lactic bacteria. The second relevant peak is succinic acid (R_t_ of 13.6 min). Finally, the third abundant peak at R_t_ of 15.8 min is hydrocinnamic acid, which is the biotransformed form of cinnamic acid (present in the polyphenol-rich cinnamon bark extract as it is). This is comparable with the extracted and detected secondary metabolites in presence of the standard molecule *trans*-cinnamic acid (Figure 5A). Another important detected molecule in presence of the cinnamon bark extract is butyric acid at R_t_ of 12.2 min.

Looking at the chromatogram obtained after the analysis of *B. animalis* subsp. *lactis* broth culture (Figure 6C), the profile is similar to the one of *L. plantarum*; however, the main peaks are fewer and are also less abundant, reflecting the poor biotransformation capacity seen in the presence of the standard *trans*-cinnamic acid (Figure 5B). Indeed, the detected molecules were lactic acid (R_t_ of 8.2 min), and succinic acid (R_t_ of 13.6 min). Interestingly, there is no detection of metabolites related to polyphenols. 

Finally, testing the biotransforming capacity of the probiotics as a consortium (Figure 6D), there is a richer chromatogram compared to the single-strain results, and with higher relative abundances. Additionally, in this case, lactic and succinic acid are detected (R_t_ of 8.2 min and 13.6 min, respectively), as well as butyric acid (R_t_ of 12.2 min), and hydrocinnamic acid (R_t_ of 15.8 min). Interestingly, the peaks with a retention time of 19 min could be associated with the 2-hydroxy-1,2,3-propane tricarboxylic acid 2-methyl ester (HPCME). The molecule is reported to be a plant-derived compound with possible anticancer activity [38].

#### 3.6.3. Analysis of the Metabolites Produced after Biotransformation of the In Vitro Enzymatic Digestion of the Polyphenol-Rich Cinnamon Bark Extract

Finally, the same GC-MSD analyses were conducted in presence of the in vitro digested polyphenol-rich cinnamon bark extract. In the chromatogram corresponding to the control condition, i.e., only the ethyl-acetate extract of the digested cinnamon bark extract dissolved in mMRS medium, there is only one abundant molecule (Figure 7A), that matches with an uncharacterized branched-chain sugar. Probably, it is an aglycone released after the in vitro enzymatic digestion [39].

Looking at the peak profile obtained after the fermentation of *L. plantarum* (Figure 7B), there is only one important peak with a retention time of 8.2 min, that is associated with lactic acid. This is reasonable given the significant growth recorded on this substrate (see Figure 4). Interestingly, both the chromatograms of *B. animalis* subsp. *lactis* and the probiotic consortium reveal another metabolite (Figure 7C,D), that is valeric acid (R_t_ of 7.2 min). Small peaks, that are not integrated and associated with molecules by the NIST library of the instrument, are present in the chromatograms. However, they could be digested and fermented polyphenols.

### 3.7. Evaluation of the Effects of Polyphenol-Rich and In Vitro Digested Polyphenol-Rich Cinnamon Bark Extract on Healthy and Tumor Colorectal Cell Lines

In literature is reported that cinnamon is endowed with antineoplastic potential, inhibiting human breast, lung, and ovarian tumor cell lines, as well as leukemia cells [40]. The molecules currently identified as responsible for this effect are cinnamaldehyde, cinnamic acid, and polyphenols, which are linked to the antioxidant and anti-inflammatory capacity associated with this spice.

The polyphenol-rich cinnamon bark extract and its digested form, at 23 µg/mL and 46 µg/mL of GA equivalents, were initially administered to the healthy CCD841, and to the tumoral colorectal SW480 cell lines to assess cell viability. Both the polyphenol-rich cinnamon extract and its in vitro digested form did not affect the healthy cell line vitality, showing a vitality percentage similar to the one of the control condition (Figure 8A,B). On the contrary, an inhibition of SW480 growth was evidenced in the presence of both the polyphenol-rich cinnamon bark extract as it is and in the presence of its in vitro digested form. Moreover, the effect was found to be concentration dependent. Indeed, in the presence of the extract as it is at 23 µg/mL cell viability was 82% respect to the empty condition, while the decreasing effect was relevant (*p*-value < 0.01) at 46 µg/mL, reaching 34% of viability (Figure 8A). Effects were stronger when the cells were exposed to the in vitro digested form of the extract. Indeed, at 23 µg/mL there was a significant reduction with 69% of viability (*p*-value < 0.05), while at 46 µg/mL, residual vitality was 33% (significantly reduced with respect to the control condition, *p*-value < 0.01) (Figure 8B).

Furthermore, the two cell lines were challenged with the secondary metabolites produced by *L. plantarum*, *B. animalis* subsp. *lactis*, and by the probiotic consortium, exposed to the polyphenol-rich cinnamon bark extract and its in vitro digested form. Regarding the healthy cell line CCD841, only the metabolites produced after the exposure of the probiotic consortium to the in vitro digested cinnamon bark extract at 46 µg/mL slightly affected the cell viability (66% of vitality, *p*-value < 0.1). Notably, the viability of the tumoral SW480 was impaired by the presence of the metabolites produced after the exposure of the probiotics to the in vitro digested cinnamon bark extract at all the tested concentrations (Figure 8B). The strongest effect was detected after the fermentation of the probiotic consortium at 46 µg/mL. Indeed, SW480 viability was 37%, significantly reduced respect to the control condition (*p*-value < 0.01). 

Interestingly, the secondary metabolites produced after the exposure to the polyphenol-rich cinnamon bark extract did not show the same effect of the extract as it is.

## 4. Discussion

In this work, a new polyphenol-rich extract was obtained by hot water maceration of the *Cinnamon verum* bark. Several extraction techniques with different solvents are commonly reported in literature for the recovery of bioactive compounds from cinnamon bark [41]. In our study, to find better extraction conditions, different solvents have been employed. Among the solvents used, 70% EtOH and hot water, have provided acceptable extraction yield. However, the extract obtained using ethanol could be used in further assays due to its poor solubility; moreover, the extraction with hot water avoids the co-extraction of essential oils (e.g., cinnamaldehyde) which commonly show antibacterial activity [42,43]. For these reasons, water has been selected as extraction solvent and the obtained extract was used for further experiments.

As shown by the Folin-Ciocalteau assay and the HPLC chromatogram (Figure 2A), the extract was mainly enriched in polyphenols. Indeed, the amount of total polyphenols present in the aqueous extracts was comparable to that one observed by Cheng et al. [7] (i.e., 520 mg/g in our extract and 441 mg/g), as well as the level of catechins (77 mg/g vs. 100 mg/g). These compounds are associated with a high radical scavenging activity [44]. Nevertheless, it is well known that the health benefits associated with natural molecules or phytocomplexes are correlated to the bioavailability of the compounds after digestion. However, the bioavailability depends on bioaccessibility [23,45]. Thus, to evaluate if the bioactive molecules of the polyphenol-rich cinnamon bark extract were affected and/or if new ones were freed after the enzymatic digestion, we included an in vitro simulation of the oral, the gastric, and the small intestinal phases. To obtain comparable results, we decided to employ the standardized digestion method of Brodkorb et al. [28], which could be carried on in static models, separating the gastric to the small intestinal digestion phases. The obtained results reveal that the concentrations of total polyphenols, flavonoids and catechins decrease significantly (Table 1, Figure 2). Indeed, the enzyme activity and the pH variation during the digestive phases can determine the degradation and/or transformation of these metabolites. In particular, the alkaline pH of the intestine facilitates the degradation of several phenolic compounds [8]. In fact, after the in vitro digestion process, total polyphenols, flavonoids, and catechins decrease by 57.3%, 98.9%, 99.5%, respectively, whereas the cinnamic acid remains the main compound.

Therefore, due to the composition of the polyphenol-rich cinnamon bark extract and its digested form, we decided to evaluate the biological activities considering both beneficial effects on bacteria and host. The potential antimicrobial activity was initially evaluated, taking into account Gram-positive and -negative microorganisms since polyphenols have antimicrobial properties [46,47]. In particular, flavonoids display high antimicrobial potential because they are synthesized by the plants after a microbial challenge [47]. Curiously, the not digested extract was active only against *S. aureus* and *B. subtilis*, at a MIC of 2 mg/mL, and 1.3 mg/mL, respectively. This is in line with literature data, reporting that spices could act differentially, with a more powerful antimicrobial potential against Gram-positive bacteria with respect to the negative ones. Probably, this is for the diverse composition and morphology of the external cell membranes. Such effect, evaluated as inhibition halo, was observed in presence of an aqueous cinnamon extract rich in alkaloids, saponins, tannins, flavonoids, steroids, and terpenoids [48]. Nevertheless, the antimicrobial activity depends also on the kind of extraction. Nabavi et al. [4] evidenced the activity of a hydro-distilled cinnamon bark essential oil against six different bacterial strains. Among them, there are also *S. aureus* and *Bacillus licheniformis*. Interestingly, the reported MIC values spanned between 2.9 and 4.8 mg/mL. Therefore, our hot-water polyphenol-rich cinnamon bark extract was more effective against the Gram-positive bacteria, though the action was not the same against *E. coli*, *P. aeruginosa*, and *C. albicans*. Regarding the in vitro digested polyphenol-rich cinnamon bark extract, we observed a complete loss of antimicrobial. This is probably connected to the transformation of the molecules present within the extract, in particular to the almost complete digestion of flavonoids and catechins, which interact with the lipid bilayer of bacterial membranes, inhibiting the synthesis of several enzymes [47]. According to the obtained indications, we investigated if the polyphenol-rich cinnamon bark extract could influence the viability of the selected probiotic strains, belonging to the Gram-positive group. Indeed, the use of concentrated natural extracts and the registered antimicrobial effect against Gram-positive bacteria could antagonize the beneficial properties of the probiotic strains. As regards, Feniman et al. [49] noted that a cinnamon extract tested at 1% concentration had detrimental effects on *Lactobacillus* and *Bifidobacterium* strains, leading to the disruption of the cell wall and the cytoplasmic membrane, causing the leakage of the cells. Interestingly, at the same polyphenol-rich cinnamon extract concentration of 1% *w*/*v*, our probiotic strains did not suffer considerably, reaching the same growth values as the negative control. Unexpectedly, the growth of *L. fermentum* and *L. reuteri* was even supported (Figure 4). Nevertheless, in literature it is reported that several Lactobacilli (such as *L. fermentum*) could utilize phenolic acids thanks to decarboxylation and/or reduction reactions and this ability is strain-specific or species-specific. Moreover, Lactobacilli could tolerate the presence of these compounds much more than *Clostridium* and *Bacteroides* members [50]. The behavior is reflected in the results of growth obtained on the in vitro digested polyphenol-rich cinnamon bark extract because the growth of the bacteria was sustained. This is probably due to the release of fermentable molecules, like sugars, cinnamic acids, and related compounds. Indeed, flavonoids are commonly glycosylated with glucose and rhamnose, but also xylose, glucuronic acid, and galactose [51]. Furthermore, homo-fermentative lactic acid bacteria are known to encode for enzymes active in the metabolism of phenolic acids. For example, members of the *Lactobacillus* genus could reduce hydroxycinnamic acids thanks to the enzymes HcrB, HcrF, and Par1 [14]. The presence of such active proteins could be correlated to the detected metabolites in the broth cultures, after the fermentation of the standard molecule *trans*-cinnamic acid and polyphenol-rich cinnamon bark extract. Indeed, we observed transformed molecules related to the cinnamic acids. After the exposition to the pure cinnamic acid, only the hydrocinnamic acid was detected in the gas-chromatography analyses. This is in line with literature because it is reported that *Lactobacillus pastorianus* var. *quinicus* could reduce the cinnamic acid to dihydrocinnamic acid without other decarboxylation reactions [52]. Instead, the presence of SCFAs and BCFAs confirmed the growth of the probiotic strains on the in vitro digested extract. In literature it is reported that the most abundant short-chain fatty acids are acetic (composed of two carbon atoms, C2), propionic (C3), and butyric (C4) acids; while the BCFAs include isobutyric, isovaleric, and valeric acids [9]. Therefore, looking at the metabolites and taking into account the complexity of the involved pathways, we can speculate that phenolic compounds could be firstly biotransformed by the bacteria, leading to accumulations of metabolites similar to the original molecule. Then, the ongoing fermentation process leads to the final metabolites that are in contact with colonocytes [9]. Therefore, we evaluated the effects of the metabolites released by the probiotics, considering the non-fermented polyphenol-rich cinnamon bark extract and its digested form, on two colorectal cell lines: the healthy CCD841 line and the tumoral SW480 line. The results highlighted that the non-fermented samples have an impact on the colorectal tumoral cell line, decreasing its viability, while no effect was detected on the healthy one. Regarding the probiotic fermented molecules, the same inhibition of the tumoral cell line viability was observed in presence of the in vitro digested cinnamon bark extract fermented by the probiotic consortium. The effect is in line with literature data, reporting that Caco-2 cells, treated with a hot-water cinnamon extract, decreased cell proliferation after 48 h [53]. Moreover, the vitality of HT-29 cell line could be inhibited by 35–85% in presence of an aqueous cinnamon extract [54], or in presence of polyphenolic molecules [38]. Therefore, the authors speculated on the induction of the pro-apoptotic molecule activation, and NF-kB and AP1 down-regulation [54]. Interestingly, our data highlight that the antitumoral effect is effectively mediated by the action of the beneficial microbes present in our gut microbiota.

## 5. Conclusions

In conclusion, in this work a polyphenol-rich cinnamon bark extract was obtained. It possesses antimicrobial activity against the tested Gram-positive bacteria at a low concentration, but without affecting the viability of probiotic strains. The in vitro digestion of the extract produced an increase of cinnamic acid, reducing the antimicrobial activity and enhancing the growth of the probiotic strains. Furthermore, it was demonstrated that this extract showed antitumoral effects on colorectal cell lines, that are implemented in presence of the probiotic fermentation. Additionally, the digested extract exerts the antitumoral effect on SW480 cell line, which is maintained after the probiotic fermentation.

This work gives elucidations regarding new possible nutraceuticals or food constituents, considering both the enzymatic and the bacterial fermentation of the molecules present in the cinnamon extract.

## Figures and Tables

**Figure 1 foods-11-03202-f001:**
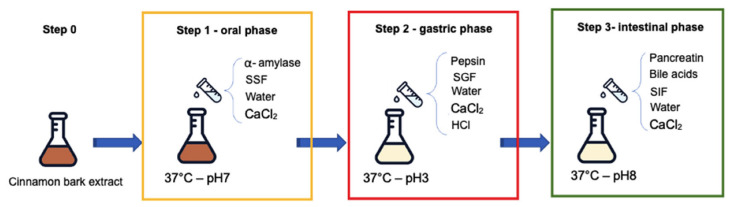
Protocol scheme of the in vitro digestion method employing INFOGEST protocol. The scheme shows the three steps of the digestion phases, including details regarding temperature, pH, buffers, and involved enzymes for each step. SSF: simulated salivary fluid; SGF: simulated gastric fluid; SIF: simulated intestinal fluid.

**Figure 2 foods-11-03202-f002:**
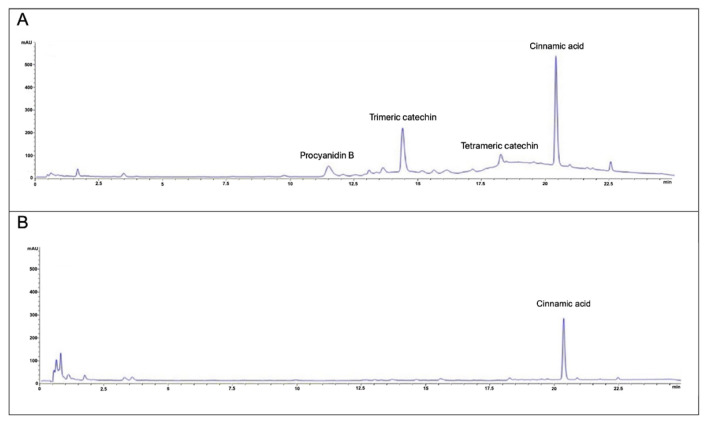
Chromatographic profile of the polyphenol-rich cinnamon bark extract before and after in vitro digestion. The figure represents the chromatograms obtained through HPLC analysis, before (**A**) and after (**B**) in vitro enzymatic digestion.

**Figure 3 foods-11-03202-f003:**
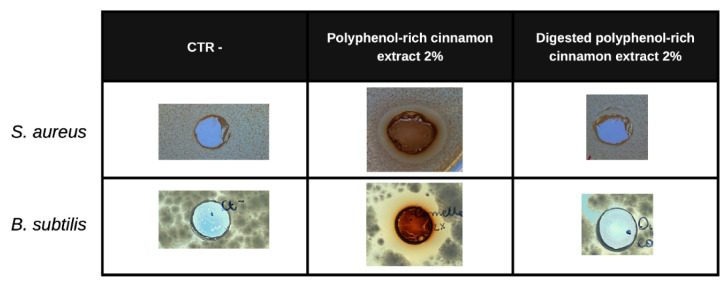
Antimicrobial activity of polyphenol-rich cinnamon bark extract before and after in vitro digestion. The figure represents only the positive results of the AWDA conducted with the polyphenol-rich cinnamon bark extract and the in vitro digested polyphenol-rich cinnamon bark extract (after oral, gastric, and intestinal phases) at 2% *w*/*v*. The positive results are growth inhibition haloes, as depicted in the column “polyphenol-rich cinnamon extract 2%”, indicating the sensitivity of *S. aureus* and *B. subtilis* to the natural extract. “CTR-” is the negative control showing no inhibition halo; “digested polyphenol-rich cinnamon extract 2%” is the digested form of the extract, showing no inhibition halo.

**Figure 4 foods-11-03202-f004:**
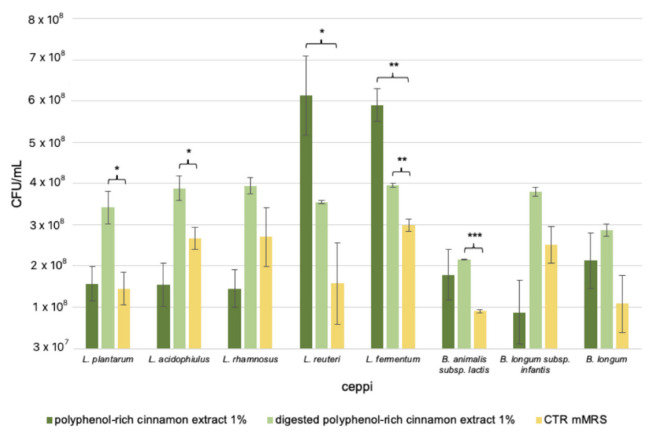
Analysis of the probiotic potential of polyphenol-rich and digested cinnamon bark extract on *Lactobacillus* and *Bifidobacterium* strains. Growth levels of the selected probiotic strains in the presence of polyphenol-rich cinnamon bark extract, digested polyphenol-rich cinnamon bark extract, and CTR medium at 1% concentration. Values are represented as mean values of CFU/mL ± SE. Statistical differences were calculated using *t*-Student’s test: * *p*-value < 0.1, ** < 0.05, and *** < 0.01.

**Figure 5 foods-11-03202-f005:**
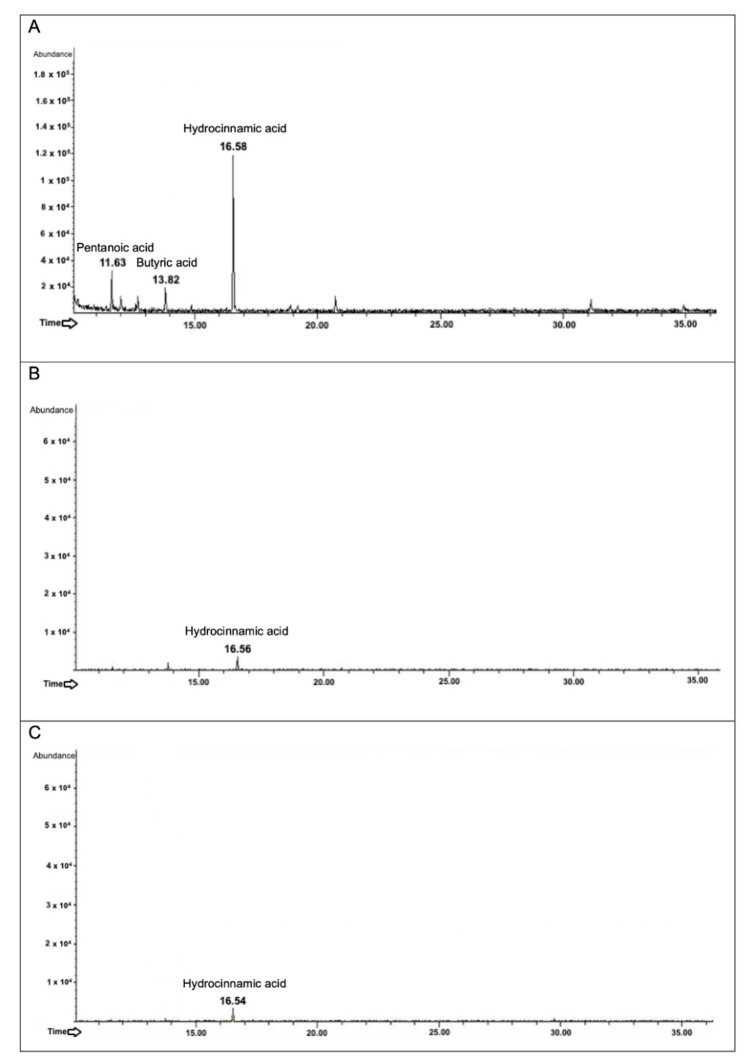
Secondary metabolites produced after the biotransformation of *trans*-cinnamic acid by *L. plantarum*, *B. animalis* subsp. *lactis*, and probiotic consortium. The figure shows the GC-MSD chromatograms. *L. plantarum*, *B. animalis* subsp. *lactis*, and the probiotic consortium were exposed for 48 h to *trans*-cinnamic acid at a final concentration of 1 mM. The identified intermediates are reported in (**A**) for *L. plantarum*, (**B**) for *B. animalis* subsp. *lactis*, and (**C**) for the probiotic consortium.

**Figure 6 foods-11-03202-f006:**
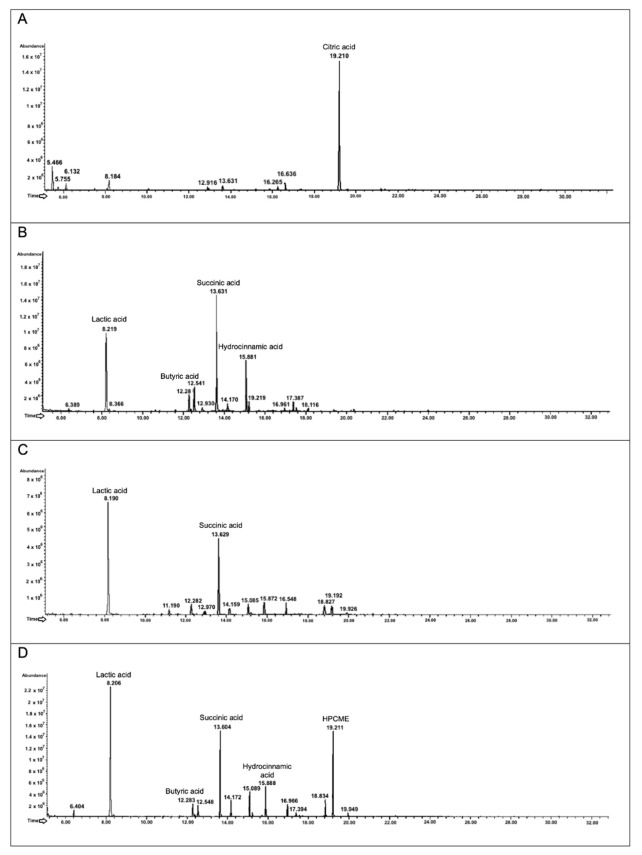
Secondary metabolites produced after the growth in presence of polyphenol-rich cinnamon bark extract of *L. plantarum*, *B. animalis* subsp. *lactis*, and probiotic consortium. The figure represents the chromatograms obtained through GC-MSD analysis. *L. plantarum*, *B. animalis* subsp. *lactis*, and the probiotic consortium were incubated for 48 h in presence of the polyphenol-rich cinnamon bark extract (1% *w*/*v*). Panel (**A**) represents the extracted polyphenol-rich cinnamon bark extract, while the identified intermediate metabolites are reported in (**B**) for *L. plantarum*, (**C**) for *B. animalis* subsp. *lactis*, and (**D**) for the probiotic consortium.

**Figure 7 foods-11-03202-f007:**
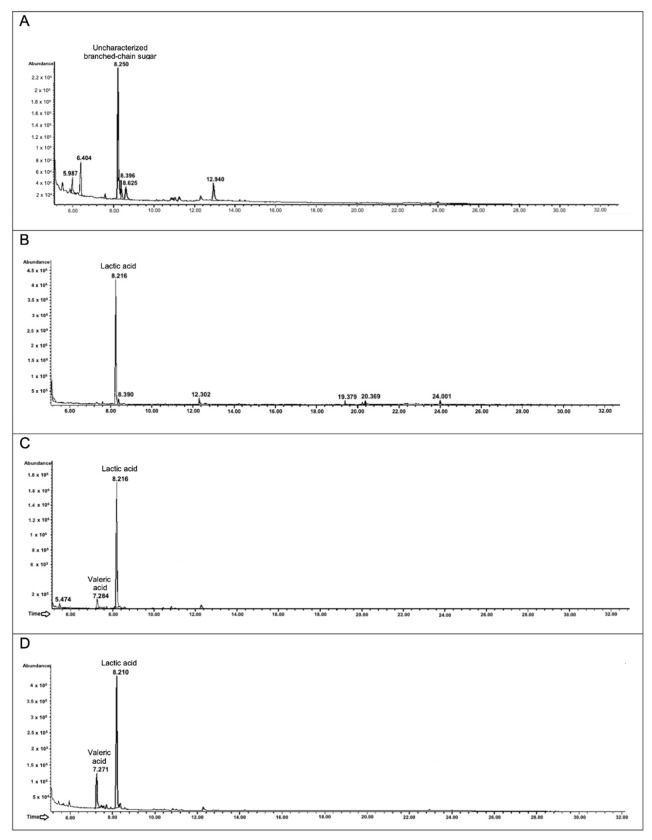
Secondary metabolites produced after the growth of *L. plantarum*, *B. animalis* subsp. *lactis*, and probiotic consortium in presence of in vitro digested polyphenol-rich cinnamon bark extract. In the figures are presented the GC-MSD profiles. *L. plantarum*, *B. animalis* subsp. *lactis*, and the probiotic consortium were incubated for 48 h in presence of the basal medium added with in vitro digested polyphenol-rich cinnamon bark extract (1% *w*/*v*). The chromatogram in (**A**) represents the extracted in vitro digested polyphenol-rich cinnamon bark extract, while the identified intermediate metabolites are reported in (**B**) for *L. plantarum*, (**C**) for *B. animalis* subsp. *lactis*, and (**D**) for the probiotic consortium.

**Figure 8 foods-11-03202-f008:**
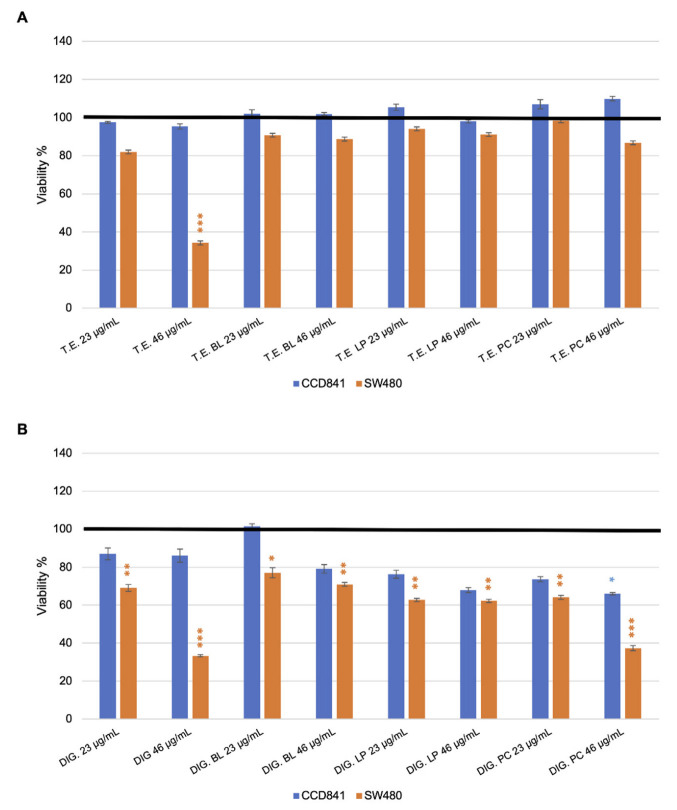
Effects on CCD841 and SW480 in the presence of polyphenol-rich cinnamon bark extract (T.E.), the secondary metabolites produced after the extract fermentation by *B. animalis* subsp. *lactis* (T.E. BL), *L*. *plantarum* (T.E. LP), and the probiotic consortium (T.E. PC) (**A**). Effects on the vitality of CCD841 and SW480 cell lines in the presence of in vitro digested polyphenol-rich cinnamon bark extract (DIG.), the secondary metabolites produced after the in vitro digested extract fermentation by *B*. *animalis* subsp. *lactis* (DIG. BL), *L*. *plantarum* (DIG. LP), and the probiotic consortium (DIG. PC) (**B**). 23 and 46 µgGAEeq/mL of polyphenols were tested. The bold black lines represent the viability of the control condition (100%). Results are depicted as % of mean vitality ± SE. Statistical differences were calculated using Dunnett’s multiple comparisons test: * *p*-value < 0.1, ** < 0.05, *** < 0.01.

**Table 1 foods-11-03202-t001:** Composition of cinnamon bark aqueous extract and after in vitro digested process used in the study.

Molecule	Aqueous Extract	Digested Extract
Total polyphenols (µgGAEeq/mg of powder extract)	520.05 ± 17.43	222.86 ± 13.71
Flavonoids (µgCATeq/mg of powder extract)	294.77 ± 19.83	3.01 ± 0.64
Catechins (µgCATeq/mg of powder extract)	77.92 ± 3.39	0.35 ± 1.88

## Data Availability

All data generated during this study are included in this article.

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
