# Peer review of "Characterization of the Biological Activities of a New Polyphenol-Rich Extract from Cinnamon Bark on a Probiotic Consortium and Its Action after Enzymatic and Microbial Fermentation on Colorectal Cell Lines"

_foods, 2022, doi:10.3390/foods11203202_

Round 1
Reviewer 1 Report
It is a very interesting job.
BACKGROUND is not part of the sections of the template to prepare manuscript. Must be INTRODUCTION.
In the text, reference numbers should be placed in square brackets [ ], and placed before the punctuation; for example [1], [1–3] or [1,3]
MATERIALS AND METHODS section
Separate the number from the units.
Change "25°C" to "25 °C".
Please check the entire manuscript.
It is suggested to use "70% EtOH" instead of “EtOH 70%"
Change “… according to Cheng et al. 2012 with same modifications.” To “… according to Cheng et al. 2012 with some modifications.”
Change “… samples were centrifuged at 600 rpm for 15 minute.” To “samples were centrifuged at 600 rpm for 15 min.”
Change "sodium hydroxide 1N" to "1 N NaOH"
Change "HCl 6 N" to "6 N HCl"
I suggest that the figure 1 caption add the meaning of SSF, SGF and SIF.
RESULTS section
Check the order of the solvents and the order of the yields of the extracted bioactive compounds. It does not agree with the statement that the hydroalcoholic extract is the one with the highest yield.
Attached file for more details.

Reviewer 2 Report
Dear author, the manuscript is very interesting, and it presents relevant information. Some changes are recommended .
King regards
-include relevant numerical results in the abstract
-Page 5- paragraph 1: indicate the solution used to adjust the pH.
-Table 1: I recommend removing this table and include the information in the text
-Page 6- paragraph 1: Reword the information of the last two sentences of this paragraph.
-Section 3.2- penultimate sentence- Indicate in the results are given in dry or wet basis
Figure 1- The information in the axes and the name of the identified compounds is not visible.
-Table 1: Indicate in the results are given in dry or wet basis
-Figure 4: Reword the Figure caption
-It is recommended to use just one “p value” to compare the results in Figure 4.
-Figure 5-7- Figures are blurred
-Figure 8: The information in the axes is not visible and the Figure caption is too extensive. -
Reviewer 3 Report
Very interesting work. I would only propose a slight revision of the English and a text reading to correct the typos.
Author Response
English language and typos have been correct.
Reviewer 4 Report
1. There are a lot of publications involving the application of polyphenols as nutraceuticals. However, the citations in the background were limited. For instance, you could refer to Foods, 2022, 11(11): 1552; Innovative Food Science & Emerging Technologies, 2022, 77(4): 102989.
2. The introduction of the phenolic metabolites should be separated to an isolated paragraph. And the authors should present more clearly why you carried out this project.
3. In 3.1, The hydroalcoholic extract, although it gave the highest EY, produced an extract with low solubility during microbiological assays conditions.
However, it seems that the EY of EtOH extract was not the highest. Please confirm the data.
4. Figure 2a, the Chromatographic profile was not clear for retention time. And the authors should further illustrate the main phenolic compounds in the extract from the HPLC results.
5. In 3.6.1, the metabolites produced after biotransformation of trans-cinnamic acid should be marked in figure 5. And it is the same for Figure 6.
6. In the Abstract, the data and results were missing. Please revise it.
Round 2
Reviewer 4 Report
The manuscript can be accepted now